# Image Restoration Using Very Deep Convolutional Encoder-Decoder Networks with Symmetric Skip Connections

**Xiao-Jiao Mao[†], Chunhua Shen[⋆], Yu-Bin Yang[†]**
[†]State Key Laboratory for Novel Software Technology, Nanjing University, China
[⋆]School of Computer Science, University of Adelaide, Australia

## Abstract

In this paper, we propose a very deep fully convolutional encoding-decoding framework for image restoration such as denoising and super-resolution. The network is composed of multiple layers of convolution and deconvolution operators, learning end-to-end mappings from corrupted images to the original ones. The convolutional layers act as the feature extractor, which capture the abstraction of image contents while eliminating noises/corruptions. Deconvolutional layers are then used to recover the image details. We propose to symmetrically link convolutional and deconvolutional layers with skip-layer connections, with which the training converges much faster and attains a higher-quality local optimum. First, the skip connections allow the signal to be back-propagated to bottom layers directly, and thus tackles the problem of gradient vanishing, making training deep networks easier and achieving restoration performance gains consequently. Second, these skip connections pass image details from convolutional layers to deconvolutional layers, which is beneficial in recovering the original image. Significantly, with the large capacity, we can handle different levels of noises using a single model. Experimental results show that our network achieves better performance than recent state-of-the-art methods.

## 1  Introduction

The task of image restoration is to recover a clean image from its corrupted observation, which is known to be an ill-posed inverse problem. By accommodating different types of corruption distributions, the same mathematical model applies to problems such as image denoising and super-resolution. Recently, deep neural networks (DNNs) have shown their superior performance in image processing and computer vision tasks, ranging from high-level recognition, semantic segmentation to low-level denoising, super-resolution, deblur, inpainting and recovering raw images from compressed ones. Despite the progress that DNNs achieve, some research questions remain to be answered. For example, can a deeper network in general achieve better performance? Can we design a single deep model which is capable to handle different levels of corruptions?

Observing recent superior performance of DNNs on image processing tasks, we propose a convolutional neural network (CNN)-based framework for image restoration. We observe that in order to obtain good restoration performance, it is beneficial to train a very deep model. Meanwhile, we show that it is possible to achieve very promising performance with a single network when processing multiple different levels of corruptions due to the benefits of large-capacity networks. Specifically, the proposed framework learns end-to-end fully convolutional mappings from corrupted images to the clean ones. The network is composed of multiple layers of convolution and deconvolution operators. As deeper networks tend to be more difficult to train, we propose to symmetrically link convolutional

and deconvolutional layers with skip-layer connections, with which the training procedure converges much faster and is more likely to attain a high-quality local optimum.

Our main contributions are summarized as follows.

- A very deep network architecture, which consists of a chain of symmetric convolutional and deconvolutional layers, for image restoration is proposed in this paper. The convolutional layers act as the feature extractor which encode the primary components of image contents while eliminating the corruption. The deconvolutional layers then decode the image abstraction to recover the image content details.
- We propose to add skip connections between corresponding convolutional and deconvolutional layers. These skip connections help to back-propagate the gradients to bottom layers and pass image details to top layers, making training of the end-to-end mapping easier and more effective, and thus achieving performance improvement while the network going deeper. Relying on the large capacity and fitting ability of our very deep network, we also propose to handle different level of noises/corruption using a single model.
- Experimental results demonstrate the advantages of our network over other recent state-of-the-art methods on image denoising and super-resolution, setting new records on these topics.[1]

**Related work** Extensive work has been done on image restoration in the literature. See detail reviews in a survey [21]. Traditional methods such as Total variation [24, 23], BM3D algorithm [5] and dictionary learning based methods [31, 10, 2] have shown very good performance on image restoration topics such as image denoising and super-resolution. Since image restoration is in general an ill-posed problem, the use of regularization [34, 9] has been proved to be essential.

An active and probably more promising category for image restoration is the DNN based methods. Stacked denoising auto-encoder [29] is one of the most well-known DNN models which can be used for image restoration. Xie et al. [32] combined sparse coding and DNN pre-trained with denoising auto-encoder for low-level vision tasks such as image denoising and inpainting. Other neural networks based methods such as multi-layer perceptron [1] and CNN [15] for image denoising, as well as DNN for image or video super-resolution [4, 30, 7, 14] and compression artifacts reduction [6] have been actively studied in these years.

Burger et al. [1] presented a patch-based algorithm learned with a plain multi-layer perceptron. They also concluded that with large networks, large training data, neural networks can achieve state-of-the-art image denoising performance. Jain and Seung [15] proposed a fully convolutional CNN for denoising. They found that CNNs provide comparable or even superior performance to wavelet and Markov Random Field (MRF) methods. Cui et al. [4] employed non-local self-similarity (NLSS) search on the input image in multi-scale, and then used collaborative local auto-encoder for super-resolution in a layer by layer fashion. Dong et al. [7] proposed to directly learn an end-to-end mapping between the low/high-resolution images. Wang et al. [30] argued that domain expertise represented by the conventional sparse coding can be combined to achieve further improved results. An advantage of DNN methods is that these methods are purely data driven and no assumptions about the noise distributions are made.

## 2   Very deep RED-Net for Image Restoration

The proposed framework mainly contains a chain of convolutional layers and symmetric deconvolutional layers, as shown in Figure 1. We term our method "RED-Net"—very deep Residual Encoder-Decoder Networks.

### 2.1   Architecture

The framework is fully convolutional and deconvolutional. Rectification layers are added after each convolution and deconvolution. The convolutional layers act as feature extractor, which preserve the primary components of objects in the image and meanwhile eliminating the corruptions. The deconvolutional layers are then combined to recover the details of image contents. The output of the deconvolutional layers is the "clean" version of the input image. Moreover, skip connections

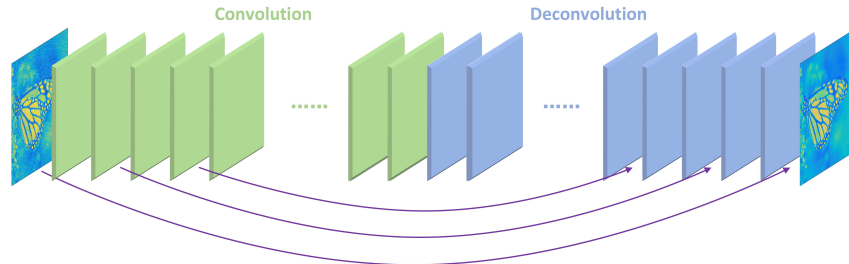

**Figure 1:** The overall architecture of our proposed network. The network contains layers of symmetric convolution (encoder) and deconvolution (decoder). Skip-layer connections are connected every a few (in our experiments, two) layers.

are also added from a convolutional layer to its corresponding mirrored deconvolutional layer. The convolutional feature maps are passed to and summed with the deconvolutional feature maps element-wise, and passed to the next layer after rectification.

For low-level image restoration problems, we prefer using neither pooling nor unpooling in the network as usually pooling discards useful image details that are essential for these tasks. Motivated by the VGG model [27], the kernel size for convolution and deconvolution is set to 3×3, which has shown excellent image recognition performance. It is worth mentioning that the size of input image can be arbitrary since our network is essentially a pixel-wise prediction. The input and output of the network are images of the same size $w \times h \times c$, where $w$, $h$ and $c$ are width, height and number of channels. In this paper, we use $c = 1$ although it is straightforward to apply to images with $c > 1$. We found that using 64 feature maps for convolutional and deconvolutional layers achieves satisfactory results, although more feature maps leads to slightly better performance. Deriving from the above architecture, in this work we mainly conduct experiments with two networks, which are 20-layer and 30-layer respectively.

### 2.1.1 Deconvolution decoder

Architectures combining layers of convolution and deconvolution [22, 12] have been proposed for semantic segmentation lately. In contrast to convolutional layers, in which multiple input activations within a filter window are fused to output a single activation, deconvolutional layers associate a single input activation with multiple outputs. Deconvolution is usually used as *learnable up-sampling layers*. One can simply replace deconvolution with convolution, which results in an architecture that is very similar to recently proposed very deep fully convolutional neural networks [19, 7]. However, there exist differences between a fully convolution model and our model.

First, in the fully convolution case, the noise is eliminated step by step, i.e., the noise level is reduced after each layer. During this process, the details of the image content may be lost. Nevertheless, in our network, convolution preserves the primary image content. Then deconvolution is used to compensate the details. We compare the 5-layer and 10-layer fully convolutional network with our network (combining convolution and deconvolution, but without skip connection). For fully convolutional networks, we use padding and up-sample the input to make the input and output the same size. For our network, the first 5 layers are convolutional and the second 5 layers are deconvolutional. All the other parameters for training are the same. In terms of Peak Signal-to-Noise Ratio (PSNR), using deconvolution works slightly better than the fully convolutional counterpart.

On the other hand, to apply deep learning models on devices with limited computing power such as mobile phones, one has to speed-up the testing phase. In this situation, we propose to use down-sampling in convolutional layers to reduce the size of the feature maps. In order to obtain an output of the same size as the input, deconvolution is used to up-sample the feature maps in the symmetric deconvolutional layers. We experimentally found that the testing efficiency can be well improved with almost negligible performance degradation.

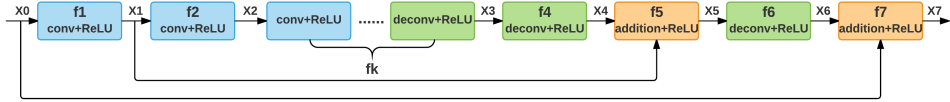

**Figure 2:** An example of a building block in the proposed framework. For ease of visualization, only two skip connections are shown in this example, and the ones in layers represented by $f_k$ are omitted.

### 2.1.2 Skip connections

An intuitive question is that, is deconvolution able to recover image details from the image abstraction only? We find that in shallow networks with only a few layers of convolution, deconvolution is able to recover the details. However, when the network goes deeper or using operations such as max pooling, deconvolution does not work so well, possibly because too much image detail is already lost in the convolution. The second question is that, when our network goes deeper, does it achieve performance gain? We observe that deeper networks often suffer from gradient vanishing and become hard to train—a problem that is well addressed in the literature.

To address the above two problems, inspired by highway networks [28] and deep residual networks [11], we add skip connections between two corresponding convolutional and deconvolutional layers as shown in Figure 1. A building block is shown in Figure 2. There are two reasons for using such connections. First, when the network goes deeper, as mentioned above, image details can be lost, making deconvolution weaker in recovering them. However, the feature maps passed by skip connections carry much image detail, which helps deconvolution to recover a better clean image. Second, the skip connections also achieve benefits on back-propagating the gradient to bottom layers, which makes training deeper network much easier as observed in [28] and [11]. Note that our skip layer connections are very different from the ones proposed in [28] and [11], where the only concern is on the optimization side. In our case, we want to pass information of the convolutional feature maps to the corresponding deconvolutional layers.

Instead of directly learning the mappings from input $X$ to the output $Y$, we would like the network to fit the residual [11] of the problem, which is denoted as $\mathcal{F}(X) = Y - X$. Such a learning strategy is applied to inner blocks of the encoding-decoding network to make training more effective. Skip connections are passed every two convolutional layers to their mirrored deconvolutional layers. Other configurations are possible and our experiments show that this configuration already works very well. Using such skip connections makes the network easier to be trained and gains restoration performance via increasing network depth.

The very deep highway networks [28] are essentially feed-forward long short-term memory (LSTMs) with forget gates, and the CNN layers of deep residual network [11] are feed-forward LSTMs without gates. Note that our deep residual networks are in general not in the format of standard feed-forward LSTMs.

## 2.2 Discussions

**Training with symmetric skip connections** As mentioned above, using skip connections mainly has two benefits: (1) passing image detail forwardly, which helps to recover clean images and (2) passing gradient backwardly, which helps to find better local minimum. We design experiments to show these observations.

We first compare two networks trained for denoising noises of $\sigma = 70$. In the first network, we use 5 layers of 3×3 convolution with stride 3. The input size of training data is 243×243, which results in a vector after 5 layers of convolution. Then deconvolution is used to recover the input. The second network uses the same settings as the first one, except for adding skip connections. The results are show in Figure 3(a). We can observe that it is hard for deconvolution to recover details from only a vector encoding the abstraction of the input, which shows that the ability on recovering image details for deconvolution is limited. However, if we use skip connections, the network can still recover the input, because details are passed from top layers by skip connections.

We also train five networks to show that using skip connections help to back-propagate gradient in training to better fit the end-to-end mapping, as shown in Figure 3(b). The five networks are: 10, 20 and 30 layer networks without skip connections, and 20, 30 layer networks with skip connections.

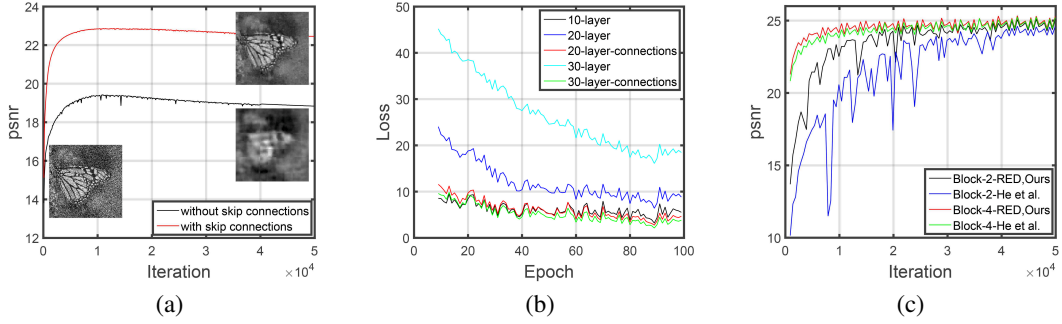

**Figure 3:** Analysis on skip connections: (a) Recovering image details using deconvolution and skip connections; (b) The training loss during training; (c) Comparisons of skip connection types in [11] and our model, where "Block-$i$-RED" is the connections in our model with block size $i$ and "Block-$i$-He et al." is the connections in He et al. [11] with block size $i$. PSNR values at the last iteration for the curves are: 25.08, 24.59, 25.30 and 25.21.

As we can see, the training loss increases when the network going deeper without skip connections (similar phenomenon is also observed in [11]), but we obtain a lower loss value when using them.

**Comparison with deep residual networks [11]** One may use different types of skip connections in our network, a straightforward alternate is that in [11]. In [11], the skip connections are added to divide the network into sequential blocks. A benefit of our model is that our skip connections have element-wise correspondence, which can be very important in pixel-wise prediction problems. We carry out experiments to compare the two types of skip connections. Here the block size indicates the span of the connections. The results are shown in Figure 3(c). We can observe that our connections often converge to a better optimum, demonstrating that element-wise correspondence can be important.

**Dealing with different levels of noises/corruption** An important question is that, can we handle different levels of corruption with a single model? Almost all existing methods need to train different models for different levels of corruptions. Typically these methods need to estimate the corruption level at first. We use a trained model in [1], to denoise different levels of noises with $\sigma$ being 10, 30, 50 and 70. The obtained average PSNR on the 14 images are 29.95dB, 27.81dB, 18.62dB and 14.84dB, respectively. The results show that the parameters trained on a single noise level cannot handle different levels of noises well. Therefore, in this paper, we aim to train a single model for recovering different levels of corruption, which are different noise levels in the task of image denoising and different scaling parameters in image super-resolution. The large capacity of the network is the key to this success.

## 2.3 Training

Learning the end-to-end mapping from corrupted images to clean ones needs to estimate the weights $\Theta$ represented by the convolutional and deconvolutional kernels. This is achieved by minimizing the Euclidean loss between the outputs of the network and the clean image. In specific, given a collection of $N$ training sample pairs $X_i, Y_i$, where $X_i$ is a corrupted image and $Y_i$ is the clean version as the ground-truth. We minimize the following Mean Squared Error (MSE):

$$\mathcal{L}(\Theta) = \frac{1}{N} \sum_{i=1}^{N} \|\mathcal{F}(X_i; \Theta) - Y_i\|_F^2. \tag{1}$$

We implement and train our network using Caffe [16]. In practice, we find that using Adam [17] with learning rate $10^{-4}$ for training converges faster than using traditional stochastic gradient descent (SGD). The base learning rate for all layers are the same, different from [7, 15], in which a smaller learning rate is set for the last layer. This trick is not necessary in our network.

Following general settings in the literature, we use gray-scale image for denoising and the luminance channel for super-resolution in this paper. 300 images from the Berkeley Segmentation Dataset (BSD) [20] are used to generate the training set. For each image, patches of size $50\times50$ are sampled

as ground-truth. For denoising, we add additive Gaussian noise to the patches multiple times to generate a large training set (about 0.5M). For super-resolution, we first down-sample a patch and then up-sample it to its original size, obtaining a low-resolution version as the input of the network.

## 2.4 Testing

Although trained on local patches, our network can perform denoising and super-resolution on images of arbitrary size. Given a testing image, one can simply go forward through the network, which is able to obtain a better performance than existing methods. To achieve smoother results, we propose to process a corrupted image on multiple orientations. Different from segmentation, the filter kernels in our network only eliminate the corruptions, which is not sensitive to the orientation of image contents. Therefore, we can rotate and mirror flip the kernels and perform forward multiple times, and then average the output to obtain a smoother image. We see that this can lead to slightly better denoising and super-resolution performance.

# 3 Experiments

In this section, we provide evaluation of denoising and super-resolution performance of our models against a few existing state-of-the-art methods. Denoising experiments are performed on two datasets: 14 common benchmark images [33, 3, 18, 9] and the BSD200 dataset. We test additive Gaussian noises with zero mean and standard deviation $\sigma = 10, 30, 50$ and 70 respectively. BM3D [5], NCSR [8], EPLL [34], PCLR [3], PDPD [33] and WMMN [9] are compared with our method. For super-resolution, we compare our network with SRCNN [7], NBSRF [25], CSCN [30], CSC [10], TSE [13] and ARFL+ [26] on three datasets: Set5, Set14 and BSD100. The scaling parameter is tested with 2, 3 and 4.

Peak Signal-to-Noise Ratio (PSNR) and Structural SIMilarity (SSIM) index are calculated for evaluation. For our method, which is denoted as RED-Net, we implement three versions: RED10 contains 5 convolutional and deconvolutional layers without skip connections, RED20 contains 10 convolutional and deconvolutional layers with skip connections, and RED30 contains 15 convolutional and deconvolutional layers with skip connections.

## 3.1 Image Denoising

**Evaluation on the 14 images** Table 1 presents the PSNR and SSIM results of $\sigma$ 10, 30, 50, and 70. We can make some observations from the results. First of all, the 10 layer convolutional and deconvolutional network has already achieved better results than the state-of-the-art methods, which demonstrates that combining convolution and deconvolution for denoising works well, even without any skip connections. Moreover, when the network goes deeper, the skip connections proposed in this paper help to achieve even better denoising performance, which exceeds the existing best method WNNM [9] by 0.32dB, 0.43dB, 0.49dB and 0.51dB on noise levels of $\sigma$ being 10, 30, 50 and 70 respectively. While WNNM is only slightly better than the second best existing method PCLR [3] by 0.01dB, 0.06dB, 0.03dB and 0.01dB respectively, which shows the large improvement of our model. Last, we can observe that the more complex the noise is, the more improvement our model achieves than other methods. Similar observations can be made on the evaluation of SSIM.

**Table 1:** Average PSNR and SSIM results of $\sigma$ 10, 30, 50, 70 for the 14 images.

| | BM3D | EPLL | NCSR | PCLR | PGPD | WNNM | RED10 | RED20 | RED30 |
|---|---|---|---|---|---|---|---|---|---|
| | | | | | PSNR | | | | |
| $\sigma = 10$ | 34.18 | 33.98 | 34.27 | 34.48 | 34.22 | 34.49 | 34.62 | 34.74 | **34.81** |
| $\sigma = 30$ | 28.49 | 28.35 | 28.44 | 28.68 | 28.55 | 28.74 | 28.95 | 29.10 | **29.17** |
| $\sigma = 50$ | 26.08 | 25.97 | 25.93 | 26.29 | 26.19 | 26.32 | 26.51 | 26.72 | **26.81** |
| $\sigma = 70$ | 24.65 | 24.47 | 24.36 | 24.79 | 24.71 | 24.80 | 24.97 | 25.23 | **25.31** |
| | | | | | SSIM | | | | |
| $\sigma = 10$ | 0.9339 | 0.9332 | 0.9342 | 0.9366 | 0.9309 | 0.9363 | 0.9374 | 0.9392 | **0.9402** |
| $\sigma = 30$ | 0.8204 | 0.8200 | 0.8203 | 0.8263 | 0.8199 | 0.8273 | 0.8327 | 0.8396 | **0.8423** |
| $\sigma = 50$ | 0.7427 | 0.7354 | 0.7415 | 0.7538 | 0.7442 | 0.7517 | 0.7571 | 0.7689 | **0.7733** |
| $\sigma = 70$ | 0.6882 | 0.6712 | 0.6871 | 0.6997 | 0.6913 | 0.6975 | 0.7012 | 0.7177 | **0.7206** |

**Evaluation on BSD200** For testing efficiency, we convert the images to gray-scale and resize them to smaller ones on BSD-200. Then all the methods are run on these images to get average PSNR and SSIM results of $\sigma$ 10, 30, 50, and 70, as shown in Table 2. For existing methods, their denoising performance does not differ much, while our model achieves 0.38dB, 0.47dB, 0.49dB and 0.42dB higher of PSNR over WNNM.

Table 2: Average PSNR and SSIM results of $\sigma$ 10, 30, 50, 70 on 200 images from BSD.

| | PSNR | | | | | | | | |
|---|---|---|---|---|---|---|---|---|---|
| | BM3D | EPLL | NCSR | PCLR | PGPD | WNNM | RED10 | RED20 | RED30 |
| $\sigma = 10$ | 33.01 | 33.01 | 33.09 | 33.30 | 33.02 | 33.25 | 33.49 | 33.59 | **33.63** |
| $\sigma = 30$ | 27.31 | 27.38 | 27.23 | 27.54 | 27.33 | 27.48 | 27.79 | 27.90 | **27.95** |
| $\sigma = 50$ | 25.06 | 25.17 | 24.95 | 25.30 | 25.18 | 25.26 | 25.54 | 25.67 | **25.75** |
| $\sigma = 70$ | 23.82 | 23.81 | 23.58 | 23.94 | 23.89 | 23.95 | 24.13 | 24.33 | **24.37** |
| | SSIM | | | | | | | | |
| $\sigma = 10$ | 0.9218 | 0.9255 | 0.9226 | 0.9261 | 0.9176 | 0.9244 | 0.9290 | 0.9310 | **0.9319** |
| $\sigma = 30$ | 0.7755 | 0.7825 | 0.7738 | 0.7827 | 0.7717 | 0.7807 | 0.7918 | 0.7993 | **0.8019** |
| $\sigma = 50$ | 0.6831 | 0.6870 | 0.6777 | 0.6947 | 0.6841 | 0.6928 | 0.7032 | 0.7117 | **0.7167** |
| $\sigma = 70$ | 0.6240 | 0.6168 | 0.6166 | 0.6336 | 0.6245 | 0.6346 | 0.6367 | 0.6521 | **0.6551** |

## 3.2 Image super-resolution

The evaluation on Set5 is shown in Table 3. Our 10-layer network outperforms the compared methods already, and we achieve even better performance with deeper networks. The 30-layer network exceeds the second best method CSCN by 0.52dB, 0.56dB and 0.47dB on scales 2, 3 and 4 respectively. The evaluation on Set14 is shown in Table 4. The improvement on Set14 in not as significant as that on Set5, but we can still observe that the 30 layer network achieves higher PSNR than the second best CSCN by 0.23dB, 0.06dB and 0.1dB. The results on BSD100, as shown in Table 5, are similar to those on Set5. The second best method is still CSCN, the performance of which is worse than that of our 10 layer network. Our deeper network obtains much more performance gain than the others.

Table 3: Average PSNR and SSIM results on Set5.

| | PSNR | | | | | | | | |
|---|---|---|---|---|---|---|---|---|---|
| | SRCNN | NBSRF | CSCN | CSC | TSE | ARFL+ | RED10 | RED20 | RED30 |
| $s = 2$ | 36.66 | 36.76 | 37.14 | 36.62 | 36.50 | 36.89 | 37.43 | 37.62 | **37.66** |
| $s = 3$ | 32.75 | 32.75 | 33.26 | 32.66 | 32.62 | 32.72 | 33.43 | 33.80 | **33.82** |
| $s = 4$ | 30.49 | 30.44 | 31.04 | 30.36 | 30.33 | 30.35 | 31.12 | 31.40 | **31.51** |
| | SSIM | | | | | | | | |
| $s = 2$ | 0.9542 | 0.9552 | 0.9567 | 0.9549 | 0.9537 | 0.9559 | 0.9590 | 0.9597 | **0.9599** |
| $s = 3$ | 0.9090 | 0.9104 | 0.9167 | 0.9098 | 0.9094 | 0.9094 | 0.9197 | 0.9229 | **0.9230** |
| $s = 4$ | 0.8628 | 0.8632 | 0.8775 | 0.8607 | 0.8623 | 0.8583 | 0.8794 | 0.8847 | **0.8869** |

Table 4: Average PSNR and SSIM results on Set14.

| | PSNR | | | | | | | | |
|---|---|---|---|---|---|---|---|---|---|
| | SRCNN | NBSRF | CSCN | CSC | TSE | ARFL+ | RED10 | RED20 | RED30 |
| $s = 2$ | 32.45 | 32.45 | 32.71 | 32.31 | 32.23 | 32.52 | 32.77 | 32.87 | **32.94** |
| $s = 3$ | 29.30 | 29.25 | 29.55 | 29.15 | 29.16 | 29.23 | 29.42 | 29.61 | **29.61** |
| $s = 4$ | 27.50 | 27.42 | 27.76 | 27.30 | 27.40 | 27.41 | 27.58 | 27.80 | **27.86** |
| | SSIM | | | | | | | | |
| $s = 2$ | 0.9067 | 0.9071 | 0.9095 | 0.9070 | 0.9036 | 0.9074 | 0.9125 | 0.9138 | **0.9144** |
| $s = 3$ | 0.8215 | 0.8212 | 0.8271 | 0.8208 | 0.8197 | 0.8201 | 0.8318 | 0.8343 | **0.8341** |
| $s = 4$ | 0.7513 | 0.7511 | 0.7620 | 0.7499 | 0.7518 | 0.7483 | 0.7654 | 0.7697 | **0.7718** |

## 3.3 Evaluation using a single model

To construct the training set, we extract image patches with different noise levels and scaling parameters for denoising and super-resolution. Then a 30-layer network is trained for the two tasks respectively. The evaluation results are shown in Table 6 and Table 7. Although training with different levels of corruption, we can observe that the performance of our network only slightly degrades

**Table 5:** Average PSNR and SSIM results on BSD100 for super-resolution.

| | PSNR | | | | | | | | |
|---|---|---|---|---|---|---|---|---|---|
| | SRCNN | NBSRF | CSCN | CSC | TSE | ARFL+ | RED10 | RED20 | RED30 |
| $s = 2$ | 31.36 | 31.30 | 31.54 | 31.27 | 31.18 | 31.35 | 31.85 | 31.95 | **31.99** |
| $s = 3$ | 28.41 | 28.36 | 28.58 | 28.31 | 28.30 | 28.36 | 28.79 | 28.90 | **28.93** |
| $s = 4$ | 26.90 | 26.88 | 27.11 | 26.83 | 26.85 | 26.86 | 27.25 | 27.35 | **27.40** |
| | SSIM | | | | | | | | |
| $s = 2$ | 0.8879 | 0.8876 | 0.8908 | 0.8876 | 0.8855 | 0.8885 | 0.8953 | 0.8969 | **0.8974** |
| $s = 3$ | 0.7863 | 0.7856 | 0.7910 | 0.7853 | 0.7843 | 0.7851 | 0.7975 | 0.7993 | **0.7994** |
| $s = 4$ | 0.7103 | 0.7110 | 0.7191 | 0.7101 | 0.7108 | 0.7091 | 0.7238 | 0.7268 | **0.7290** |

comparing to the case in which using separate models for denoising and super-resolution. This may due to the fact that the network has to fit much more complex mappings. Except that CSCN works slightly better on Set14 super-resolution with scales 3 and 4, our network still beats the existing methods, showing that our network works much better in image denoising and super-resolution even using only one single model to deal with complex corruption.

**Table 6:** Average PSNR and SSIM results for image denoising using a single 30-layer network.

| | 14 images | | | | BSD200 | | | |
|---|---|---|---|---|---|---|---|---|
| | $\sigma = 10$ | $\sigma = 30$ | $\sigma = 50$ | $\sigma = 70$ | $\sigma = 10$ | $\sigma = 30$ | $\sigma = 50$ | $\sigma = 70$ |
| PSNR | 34.49 | 29.09 | 26.75 | 25.20 | 33.38 | 27.88 | 25.69 | 24.36 |
| SSIM | 0.9368 | 0.8414 | 0.7716 | 0.7157 | 0.9280 | 0.7980 | 0.7119 | 0.6544 |

**Table 7:** Average PSNR and SSIM results for image super-resolution using a single 30-layer network.

| | Set5 | | | Set14 | | | BSD100 | | |
|---|---|---|---|---|---|---|---|---|---|
| | $s = 2$ | $s = 3$ | $s = 4$ | $s = 2$ | $s = 3$ | $s = 4$ | $s = 2$ | $s = 3$ | $s = 4$ |
| PSNR | 37.56 | 33.70 | 31.33 | 32.81 | 29.50 | 27.72 | 31.96 | 28.88 | 27.35 |
| SSIM | 0.9595 | 0.9222 | 0.8847 | 0.9135 | 0.8334 | 0.7698 | 0.8972 | 0.7993 | 0.7276 |

## 4   Conclusions

In this paper we have proposed a deep encoding and decoding framework for image restoration. Convolution and deconvolution are combined, modeling the restoration problem by extracting primary image content and recovering details. More importantly, we propose to use skip connections, which helps on recovering clean images and tackles the optimization difficulty caused by gradient vanishing, and thus obtains performance gains when the network goes deeper. Experimental results and our analysis show that our network achieves better performance than state-of-the-art methods on image denoising and super-resolution.

*This work was in part supported by Natural Science Foundation of China (Grants 61673204, 61273257, 61321491), Program for Distinguished Talents of Jiangsu Province, China (Grant 2013-XXRJ-018), Fundamental Research Funds for the Central Universities (Grant 020214380026), and Australian Research Council Future Fellowship (FT120100969). X.-J. Mao's contribution was made when visiting University of Adelaide. His visit was supported by the joint PhD program of China Scholarship Council.*

## Footnotes

[1]We have released the evaluation code at `https://bitbucket.org/chhshen/image-denoising/`

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
