[Reviews · NeurIPS 2016]

Reviewer 1

Summary

Image Restoration using Very Deep Convolutional Encoder-Decoder Networks with Symmetric Skip Connections This paper proposes to tackle the problem of image restoration and image super-resolution by using a neural net based on a convolution-deconvolution architecture with skip connections between these two stages. Contributions are multiple: 1) symmetric convolution and deconvolution layers, 2) skip connection linking symmetric convolution and deconvolution layers, to ease back-propagation and reuse otherwise lost details during deconvolution, 3) practical advantages when using deeper networks, such as independence of training the model separately with several noise/downsampling factors, and better performance than state-of-the-art. Results are impressive and shown on image restoration and image super-resolution, with comparison with several state-of-the-art methods.

Qualitative Assessment

The proposed framework indicates that using a convolution-deconvolution architecture on deep neural networks is improved when adding skip connections between symmetric encoding-decoding layers. The main contributions resides on the proposed symmetric skip connections - however - the results focus on evaluating their overall performance, while one may be interested in which skip connections contribute most in the overall performance. For instance, does the skip connections between outmost symmetric layers contribute better than skip connections between innermost symmetric layers? This may help in understanding if one should connect every second layers, all layers, or only first-last or middle layers. Minimal details on the deconvolution layers may benefit to their comprehension - it is now deferred on fully convolutional neural net papers In the super-resolution experiments, how are images downsampled and upsampled? (average/median/bilinear or else) Would you anticipate the network to be robust on the choice of the downsampling scheme? For consistency, should the wording "shortcuts" be changed for "skip connections"?

Confidence in this Review

2-Confident (read it all; understood it all reasonably well)


Reviewer 2

Summary

This paper proposed a deep fully convolutional neural network for image restoration such as denoising and super-resolution. The network is composed of multiple layers of convolution and de-convolution operators, learning end-to-end mappings from corrupted images to the ground-truth. In addition, symmetrical links are used to connect the corresponding convolution and de-convolution layers. Experiments results show that the proposed method outperforms the state-of-the-arts on image denoising and super-resolution.

Qualitative Assessment

The proposed method is very related to Kim et al.’s work, but the later one is not mentioned at all. At the time of submission, the arXiv version of Kim’s paper was already online, which should be cited and discussed. Kim, Jiwon, Jung Kwon Lee, and Kyoung Mu Lee. "Accurate image super-resolution using very deep convolutional networks." arXiv preprint arXiv:1511.04587 (2015). Accepted at CVPR 2016. In Kim et al.’s work, the model is with multiple flat convolution layers. The input is connected with the output to form a residual learning. Compared with Kim’s model, half of convolutional layers are replaced with de-convolution layers in the proposed model and it has more skip connections between the layers. It seems that the overall idea is very similar to Kim’s paper that using very deep fully convolutional layers and residual learning for image restoration. Overall, there is some advancement made by the paper, but may be not significantly enough for a NIPS paper. 1. It is claimed that this paper different corruptions using a single model. While it provides some contribution in denoising, the strategy of mixed training set has already used in Kim’s paper for different levels of super-resolution. Therefore, the contribution is not that significant. 2. In section 2.2, to validate the benefits of skip connections, it is not very convincing by comparing residual networks [11] since [11] was proposed for recognition. A more proper choice could be the version of removing the connections in the middle, with only one input-to-end connection. It is like the residual learning in Kim’s work. This experiment can validate the necessity of the middle connections. 3. I think it is weird for a paper on image restoration without any visual comparisons in the paper. although these results are included in the supplementary material. For some results, the improvements are not very clear. Highlighting some zoomed-in regions would be helpful. Minor: 1. Tables 1,2,3 and 6,7 have the same captions, but actually they are trained with different strategies. The captions should be changed to make such difference clear without reading the text in the paper. 2. In Fig. 3(a), why the PSNR decreased when the number of epochs increased? 3. Another work using symmetric fully convolution and de-convolution layers may be related to this paper. Edgar Simo-Serra et al., Learning to Simplify: Fully Convolutional Networks for Rough Sketch Cleanup. SIGGRAPH 2016. __________________________________________________________________ Update: After reading the authors’ rebuttal, I still have several concerns about this paper. The authors said that the difference between this paper and Kim’s paper is that they used more skip connections instead of a single one from the input to the output. To me, the results are not necessarily getting better by doing this. In the paper, there is no comparison between a single connection and multiple connections. Therefore, it is hard to evaluate the benefit of this. The authors only compared the multiple connections with residual learning [11], which was a different structure with a skip connection in each small block. Another import issue of this paper is that Kim’s work is never mentioned. It could mislead the readers: it seems that all the ideas of this paper (e.g., skip connection, very deep structure, handle different noise/scale by mixing the training data) are original, but actually very similar to Kim’s work. The arXiv version of Kim’s paper was online in 2015. I am not asking the authors to compare with an arXiv paper, but the difference should be discussed since the ideas are so similar. The results are also not very convincing. In the supplementary results, it is hardly to see any visual improvements over existing methods. While the “numbers” (e.g. PSNR) are slighted improved by 0.1-0.2dB, it does not mean a necessary improvement. For example, in Johnson’s paper, it shows a lower PSNR gives perceptually better looking result. I agree with the authors that this paper has slightly higher PSNR than Kim's(around 0.15dB) on Set5 and BSD 100, but lower by similar amounts on Set14. Johnson, Justin, Alexandre Alahi, and Li Fei-Fei. "Perceptual losses for real-time style transfer and super-resolution." arXiv preprint arXiv:1603.08155 (ECCV) 2016. Overall, I feel the major issue of this paper is the lack of proper discussions and evaluations.

Confidence in this Review

3-Expert (read the paper in detail, know the area, quite certain of my opinion)


Reviewer 3

Summary

This paper present an end-to-end CNN for image restoration. The key idea is to use skip connections to pass the detailed low-level image information to the later layers toward the end for reconstruction. The idea is very simple but yet very effective.

Qualitative Assessment

The paper was easy to read. The problem of image restoration is very well-studied problem. The experiments and the evaluation metrics are standard. The main idea in this paper is interesting and it completely makes sense. The reason that I am not voting very high for this paper is that the method is not very efficient compare to current efficient techniques on CPU [1] and the improvement over the CPU based methods is not very high. Efficiency of image restoration is a very important concern in general. [1] Filter forests for learning data-dependent convolutional kernels. S Ryan Fanello et al. CVPR2014

Confidence in this Review

2-Confident (read it all; understood it all reasonably well)


Reviewer 4

Summary

The paper 'Image Restoration Using Very Deep Convolutional Encoder-Decoder Networks with Symmetric Skip Connections' shows how to apply the recently popularised Residual Networks framework to the problem of image restoration. It shows state of the art performance for both, image super-resolution and image denoising. Additionally to having specialised networks for each task and each scaling factor/noise level they also show that a single network can be trained on both tasks and multiple noise levels and up-scaling factors and still show very good performance.

Qualitative Assessment

I think this paper nicely shows how to use the residual network idea for image restoration. The presentation is clear and the results are to my knowledge state of the art. I find the result of training a single network for both tasks and multiple noise levels and scaling factors particularly compelling. However, the paper could be improved by discussing the perceptual relevance of their quantitative advances. All in all a good paper that pushes the boundaries of image denoising and super-resolution. As a side note: the authors might want to include the comparison to two papers for super-resolution using Deep Neural Networks that appeared in CVPR this year (after the NIPS submission deadline): Deeply-Recursive Convolutional Network for Image Super-Resolution. Accurate Image Super-Resolution Using Very Deep Convolutional Networks. both by Jiwon Kim, Jung Kwon Lee, Kyoung Mu Lee.

Confidence in this Review

2-Confident (read it all; understood it all reasonably well)


Reviewer 5

Summary

This paper proposes a encoder-decoder framework for the task of Image restoration. The encoder proposed consists of a stack of Convolution operations and the decoder consists of a stack of deConvolution operations. In-addition they use skip connections between corresponding convolution and deconvolution layers that helps in propagating the image details to the deconvolution layers and also helps in increasing the speed of convergence. Their experimental results show that they beat previous state of the art methods in the tasks of image denoising and image super-resolution.

Qualitative Assessment

Image restoration is a well studied problem. Although the proposed model achieves state of the art performance on this task, it is only incremental. And I feel that qualitatively it is not any better compared to the 2nd method. The paper does introduce some new application of resNet with interesting skip connections, which are some positive aspects of the paper. I think its time to start performing human studies to analyze if the improvement in performance really means something according to the humans. If there is a way to show that the improvement in this task helps humans/machines in other important tasks, I would be more inclined to accept this paper. Some other weaknesses I found in the paper includes: 1. The paper mentions that their model can work well for a variety of image noise, but they show results only on images corrupted using Gaussian noise. Is there any particular reason for the same? 2. I can't find details on how they make the network fit the residual instead of directly learning the input - output mapping. - Is it through the use of skip connections? If so, this argument would make more sense if the skip connections exist after every layer (not every 2 layers) 3. It would have been nice if there was an ablation study on what plays the most important factor on the improvement in performance. Whether it is the number of layers or the skip connections, and how does the performance vary when the skip connections are used for every layer. 4. The paper says that almost all existing methods estimate the corruption level at first. There is a high possibility that the same is happening in the initial layers of their Residual net. If so, the only advantage is that theirs is end to end. 5. The authors mention in the Related works section that the use of regularization helps the problem of image- restoration, but they don’t use any type of regularization in their proposed model. It would be great if the authors can address these points (mainly 1, 2 and 3) in the rebuttal.

Confidence in this Review

2-Confident (read it all; understood it all reasonably well)


Reviewer 6

Summary

This paper proposes a deep neural network architecture for denoising and super-resolution. The architecture is a deep CNN, where the first half is composed of convolutions (with strides) and the second half deconvolutions (the reverse of convolution with stride). Skip connections are added and connect corresponding layers of convolution to the deconvolution, and is justified by the Residual Networks and to preserve the high frequency information. The network is evaluated by computing PSNR and SSIM on denoising and super-resolution tasks, and outperforms the previous methods that are shown in this paper. Some denoised and super-resolution images are showed in supplementary material.

Qualitative Assessment

Please define "deconvolution". From what I understand in the paper (and following reference 22), it seems to be a convolution with transposed kernel (with "full" output instead of "valid", which only affects the edge of the feature planes). When reading more of the paper, it seems you also use strides in the deconvolution. This has been used recently, in particular in generative networks, but as far as I know the vocabulary is not standard and a clarification from the start would make the paper easier to understand. The paper http://arxiv.org/abs/1511.05666 seems to be relevant for super-resolution and is not mentioned. One problem about this article is that it is not clear what is new contribution. Skip connections as shown here are already known for denoising autoencoders (for instance http://arxiv.org/abs/1504.08215). Convolutions and deconvolutions are known to work well for generative models, and denoising autoencoders have been there for a long time. Putting all together seems to provide good results but I'm not sure this is a NIPS poster (definitely a workshop). Finally, the evaluation using PSNR and SSIM may be a problem (maybe not, but at least a discussion about it would be nice). Indeed, PSNR (and to a lesser extent SSIM) is based on the L2 loss. In lots of cases, minimizing the L2 loss may not be satisfying. Indeed, it cannot deal with multimodal distributions, and produces the mean of the possible values when there is ambiguity. Therefore, it will work well in the cases where the target distribution is unimodal (ie. when the output is well defined), which is true in most of the test cases of the article, but won't work as well in harder cases. Measuring the PSNR won't show this problem since it is directly linked with the L2 distance, and as far as I know the only way to detect this kind of issues is visually looking at the images. The supplementary material shows lots of images, but comparisons with other models are only shown in the easy cases (denoising with white, uncorrelated noise and super-resolution with scale only 3). The harder cases (with structured noise) are not compared with other methods. Finally, showing even harder cases (for instance super-resolution with a larger scale), where the model starts to break, would prove that the examples are not cherry-picked. To summarize, this approach seems to work well in easy cases but, from the paper, we don't know how it behaves in harder cases. If it doesn't work as well, it should be discussed in the paper.

Confidence in this Review

2-Confident (read it all; understood it all reasonably well)